

# From data compilation to model validation: comparing three ecosystem models of the Tasman and Golden Bays, New Zealand

Vidette L. McGregor[1], Peter Horn[2], Adele Dutilloy[1], Samik Datta[1], Alice Rogers[3], Javier Porobic[4], Alistair Dunn[5] and Ian Tuck[1]

[1] Fisheries, National Institute of Water and Atmospheric Research Ltd., Wellington, New Zealand
[2] Pachyornis Science, Wellington, New Zealand
[3] School of Biological Sciences, Victoria University of Wellington, Wellington, New Zealand
[4] Oceans & Atmosphere, CSIRO, Hobart, Tasmania, Australia
[5] Ocean Environmental, Wellington, New Zealand

## ABSTRACT

The Tasman and Golden Bays (TBGB) are a semi-enclosed embayment system in New Zealand that supports numerous commercial and recreational activities. We present three ecosystem models of the TBGB ecosystem with varying levels of complexity, aimed at contributing as tools to aid in understanding this ecosystem and its responses to anthropogenic and natural pressures. We describe the process of data compilation through to model validation and analyse the importance of knowledge gaps with respect to model dynamics and results. We compare responses in all three models to historical fishing, and analyse similarities and differences in the dynamics of the three models. We assessed the most complex of the models against initialisation uncertainty and sensitivity to oceanographic variability and found it most sensitive to the latter. We recommend that scenarios relating to ecosystem dynamics of the TBGB ecosystem include sensitivities, especially oceanographic uncertainty, and compare responses across all three models where it is possible to do so.

## INTRODUCTION

End-to-end ecosystem models which can deal with bottom-up and top-down system controls have become popular for exploring scenarios involving human induced impacts including fishing and climate change (*Rose, 2012*). These models can be included as useful tools in providing a holistic approach to understanding system-wide repercussions of how we manage our marine resources (*McGregor et al., 2019*; *Smith et al., 2017*; *Stecken & Failler, 2016*).

The Tasman and Golden Bays (TBGB) is an appropriate focal area for Ecosystem Based Management as it supports a diverse range of marine, land and human activities, with economic, social and customary value (*Sustainable Seas, 2019*). The TBGB ecosystem is a

Corresponding author
Vidette L. McGregor,
vidette.mcgregor@niwa.co.nz

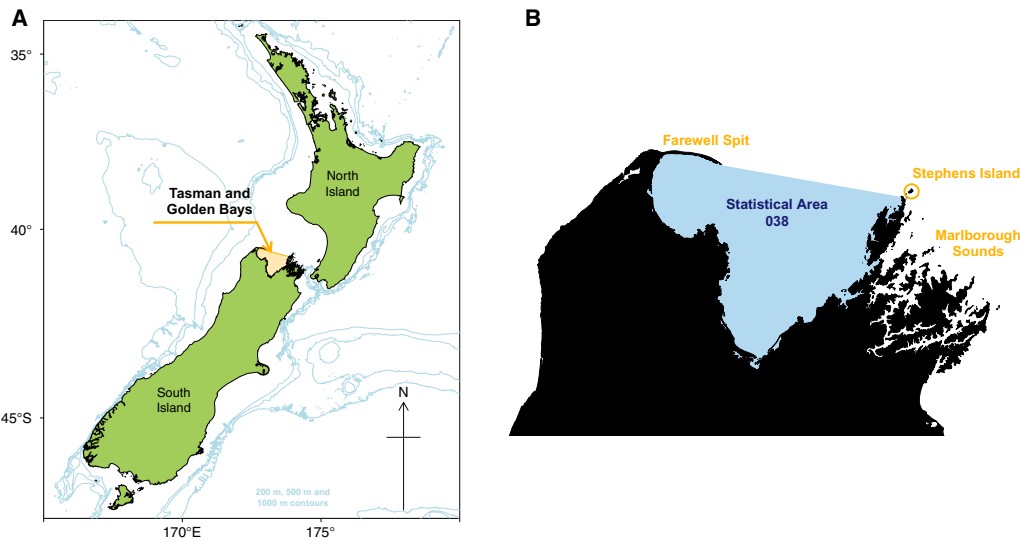

**Figure 1** Map of New Zealand with Tasman and Golden Bays marked and shaded orange, including 200 m, 500 m, and 1,000 m isobaths (A). Map of Tasman and Golden Bays with fisheries Statistical Area 038 (blue shaded) and Farewell Spit, Stephens Island and Marlborough Sounds (orange labels) (B).

relatively shallow semi-enclosed embayment system at the north of South Island, New Zealand (Fig. 1) and is the focal area for the national science challenge Sustainable Seas, aimed at developing tools and approaches for ecosystem based management. Strong ocean currents enter the system from the Tasman sea bringing with them cold, nutrient-rich waters, which make the area highly productive (*Chiswell et al., 2019*). TBGB has large sheltered areas which are home to a diverse array of habitats including large seagrass beds, rocky reefs and large sandy outcrops, which support a wide variety of species, from small reef bound species to large migrating pelagic species (*Handley, 2006*; *Stevenson & MacGibbon, 2018*).

TBGB supports numerous commercial fisheries (for finfish and invertebrates), an active recreational fishery (*Fisheries New Zealand, 2020*), and marine farming activities (*Handley, 2006*). The area is a popular destination for tourists, for example the Abel Tasman National Park is particularly popular for hiking, camping and water sports. Much of the land surrounding the bays has been modified by horticulture, forestry, and residential development.

Studies of the TBGB ecosystem include trawl surveys (*MacGibbon & Stevenson, 2013*; *Stevenson & MacGibbon, 2018*), fishery characterisations and stock assessments (*Starr & Kendrick, 2017a*, *2017b*; *Parsons et al., 2018*; *Langley, 2018*; *Williams et al., 2014*), bioregionalisation (*Handley, Dunn & Hadfield, 2018*), habitat and fishing effects (*Handley et al., 2014*), oceanography from observations and modelling (*Chiswell et al., 2019*), benthos and anthropegenic effects (*Handley, 2006*), sedimentation (*van der Linden, 1969*), and tidal circulation (*Tuckey et al., 2006*).

There are key ecological questions around historical shifts in the TBGB ecosystem. The bays used to support a large scallop fishery, but scallop recruitment has failed in recent

years (*Williams et al., 2014*, *2015*; *Tuck, Williams & Bian, 2018*). In a review of drivers of shellfish production, *Michael et al. (2015)* suggests food (primary production), suspended sediments and turbidity, changes to benthic communities and sediment, effects of fishing, and disease are all potential key drivers. There is also a large snapper fishery, which has experienced marked variations in productivity over time, and is currently producing relatively high catches (*Langley, 2018*). Snapper are known to be temperature dependent, with strong year classes found to be correlated with high sea surface temperature (SST), in particular high autumn SST (*Francis, 1993*).

We have developed three ecosystem models as part of a tool-kit for exploring and understanding the TBGB ecosystem. Each model varies in complexity and scope, model development and validation demands, and applicability to different types of questions or scenarios. Each model developed has used a different framework: TBGB_AM uses Atlantis (*Audzijonyte et al., 2017*; *Audzijonyte et al., 2019*; *Pethybridge et al., 2019*), TBGB_EwE uses Ecopath with Ecosim (EwE) (*Christensen & Walters, 2004*; *Christensen, Walters & Pauly, 2005*), and TBGB_SS is a size spectrum ecological model using the multispecies implementation of the R package *mizer* (*Scott, Blanchard & Andersen, 2014*; *Blanchard et al., 2014*).

Atlantis has been reviewed as one of the best modelling frameworks for exploring 'what-if' type questions (*Plagányi, 2007*), and has many applications globally (*Savina et al., 2005*; *Fulton, Smith & Smith, 2007*; *Link, Fulton & Gamble, 2010*; *Ainsworth, Schirripa & Morzaria-Luna, 2015*; *Smith, Fulton & Day, 2015*; *Sturludottir et al., 2018*; *Ortega-Cisneros, Cochrane & Fulton, 2017*; *McGregor et al., 2019*). As a full end-to-end ecosystem model, it is capable of including bio-physical components of an ecosystem, species functional groups, fishing fleet dynamics, social and economic dynamics (*Audzijonyte et al., 2017*, *2019*). With sufficient data, this modelling approach can be extremely useful for management strategy evaluation (*Plagányi, 2007*).

An Atlantis model is developed using components relating to the bio-physical, ecological, and human use aspects of an ecosystem. This component structure allows for additional dynamics to be added incrementally. Initially, the spatial and temporal structure are defined, and the model oceanography is forced using outputs from an ocean physics model. Initial conditions are defined for nutrients, and growth rates for bacteria, detritus and primary producers. Following this, other species functional groups can be added, which is when the complexity escalates due to many interacting dynamics such as diets, movement, habitat preferences, feedback in the microbial loop, ages, size, and trophic structure. Atlantis is a deterministic simulation model such that for a given parameter set and model specification, the model outputs are identical. Atlantis models are too complex to statistically fit to observations, so we are reliant on analysing and understanding the model dynamics to assess the suitability of the model for exploring a given scenario, and to assess the reliability of scenario outcomes. Analysing and understanding the model dynamics and potential weaknesses is essential before the model can be used to learn about the system.

EwE is trophodynamic modelling software which uses a mass-balance approach to describe ecosystem based, marine food web interactions (*Christensen & Walters, 2004*;
*Christensen, Walters & Pauly, 2005*). EwE works sequentially, where first an Ecopath model is populated and balanced at a specified point in time, then Ecosim is used to simulate the model through time (*Walters, Christensen & Pauly, 1997*). The Ecopath model is balanced by assuming that the energy removed from each species group, through fishing or predation, for example, must be balanced with the energy consumed by that group (*Christensen & Pauly, 1992*). Ecosim is then added to dynamically simulate ecosystem-based changes over time. Ecosim uses foraging arena theory (*Walters & Juanes, 1993*), which assumes only a portion of the prey biomass is available to the predators. This partitioning of prey resources can be used as a proxy for spatial dynamics, and also has a stabilizing effect on ecosystem dynamics through providing refuge to prey groups (*Walters, Christensen & Pauly, 1997*). EwE models can be used for exploring impacts of fishing in conjunction with environmental shifts or trends, and for exploring optimal fishing policies (*Christensen & Walters, 2004*). EwE models have more recently been used to produce time-series predation mortality for use in single-species stock assessments (*Bentley et al., 2019*).

The multi-species size spectrum model specifies individual traits for each species group, and also utilizes the size spectrum model for predator-prey interactions. Prey selection is a function of predator size, prey size, and the prey-species preference for a given predator. With this model, we can predict species' size distributions, abundance, productivity and predator-prey interactions. Hence, it is possible to evaluate trade-offs based on responses in community and foodweb structure, population status, diversity, and fisheries yield (*Blanchard et al., 2014*). The possible trade-offs that we can explore with this model are limited to fishing effects such as focusing fishing effort on different parts of the system, and basic environmental effects such as variations in primary productivity.

In this paper, we describe and evaluate TBGB_AM, which is the first end-to-end ecosystem model for the Tasman and Golden Bays, New Zealand, as well as the two alternative ecosystem models; TBGB_EwE and TBGB_SS. We present analyses of the models, comparing both state and dynamics to each other and to current knowledge, and make recommendations on the appropriate use of each model.

## METHODOLOGICAL APPROACH

Model development primarily focused on the Atlantis model for the TBGB ecosystem. The alternative models using the Ecopath with Ecosim (EwE) and multi-species size-structured model frameworks were developed as simplifications of TBGB_AM, and have been assessed with respect to TBGB_AM where it was possible to do so. TBGB_AM has been tested with respect to initialisation uncertainty, realised growth and mortality rates, variability from the assumed oceanographic variables, and connectivity analysis following the methods of *McGregor et al. (2019)*, and *McGregor, Fulton & Dunn (2020)*. All three models and associated R scripts are available on GitHub (*McGregor, 2019*).

The process of developing these models was not linear, but rather iterative and incremental. There were six main stages to the development, each of which was re-visited until we were satisfied with the performance of the models and our understanding of their dynamics. The main stages can be summarised as:

1. Model design: data and model inputs were collated and defined and the base TBGB_AM developed.
2. Alternative models: two alternative ecosystem models were developed; one size-structured and one Ecopath with Ecosim.
3. Calibration: the base historical TBGB_AM was calibrated without fishing such that this model had stable biomass trajectories over the 1900–2014 model period, realistic diets, growth rates, natural mortalities, with these compared to the alternative models where appropriate.
4. Sensitivity analyses: TBGB_AM was tested for sensitivity to uncertainty in the initial conditions and oceanographic variables. Simulations were explored aimed at understanding connectivity and influence between the species groups.
5. Fishing: historical fishing was included in all three models using forced catch removals.
6. Skill assessment: comparisons to abundance indices and biomass estimates were carried out for all models, including between-model comparisons.

"Model Design", "Alternative Models", "Calibration", "Sensitivity Analyses", "Fishing", "Skill Assessment" cover each of these six main stages, followed by "Bringing it Together: Comparing Modelled Ecosystem Dynamics": Bringing it together, compares the performance and dynamics as they relate to the three models.

## MODEL DESIGN

An Atlantis model is developed as components relating to the physical, biological, ecological, and fishing components of a marine ecosystem. It uses these components to simulate the ecosystem through time, calculating each new state based on the previous state and the events of the current timestep (*McGregor et al., 2019*). This section is structured similarly, and describes the physical, biological, ecological, and fishing components of TBGB_AM. Further details on Atlantis can be found in the Atlantis user manual (*Audzijonyte et al., 2017*).

### Model area

The TBGB area comprises waters in Tasman and Golden Bays, at the northern end of South Island, New Zealand (Fig. 1). The area is bounded in the north by a line connecting the eastern tip of Farewell Spit and Stephens Island at the northern extreme of the Marlborough Sounds, and by the coastal margin within the bays (but excluding estuaries and Croisilles Harbour). It equates to Statistical Area 038, one of many areas used to define the location of commercial fish catches in the New Zealand EEZ (*Mackay et al., 2005*).

An Atlantis model is defined spatially using polygons and depth layers, with each polygon/depth layer referred to as a cell. The intention of the splits is to capture important aspects of the region but at a simplified level such that modelling the region over many years becomes possible (i.e. balancing detail with computational efficiency). The modelled area was divided into 25 polygons (Fig. 2), which represent the main physical and biochemical structure of the ecosystem (*Handley, Dunn & Hadfield, 2018*) and the historical footprint of fishing activity (Fig. 2). There is one additional boundary polygon

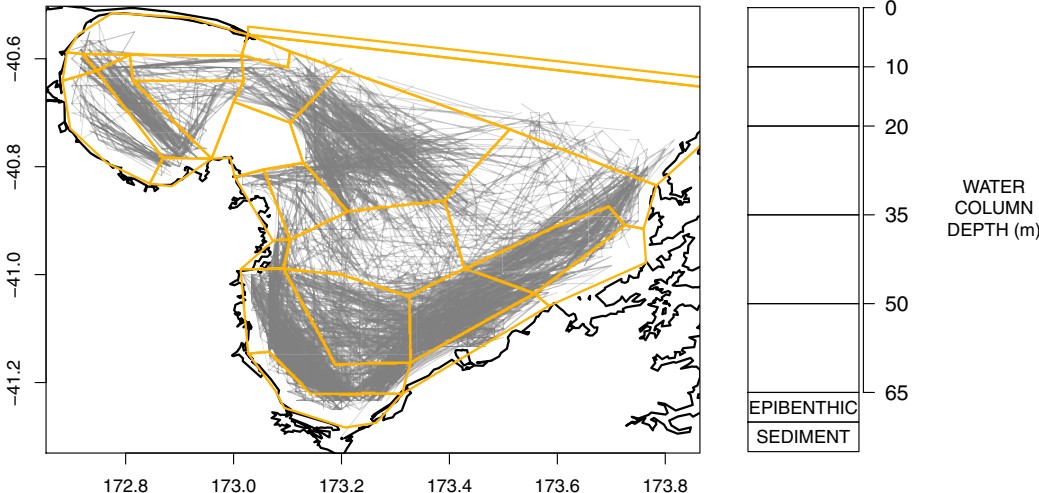

**Figure 2 Polygons as defined for TBGB_AM with historical trawl footprint (grey, left) and depth layer bins (right).**

which flanks the northern boundary and allows for the exchange of water, nutrients and biota from the dynamic model domain. All model polygons were further divided into water column depth layers, ranging from one layer in some nearshore polygons to five layers for the deepest polygons. The defined depth layers are shown in Fig. 2. In addition to the water column layers, each polygon contains one epibenthic and one sediment layer.

## Time

An Atlantis model is run with a burn-in period followed by a model period. The purpose of the burn-in period is to allow initial fluctuations resulting from the initial state of the model to stabilise (*McGregor, Fulton & Dunn, 2020*). TBGB_AM consists of a 35-year burn-in period (1865–1899), followed by a model period from 1900 to 2014 (inclusive). All results presented here are from the model period 1900–2014. The model used 12 hour timesteps to allow for changes in temperature, light and feeding patterns between night and day.

## Oceanography

Salinity, temperature and water exchange between cells were forced in TBGB_AM using outputs from a ROMS (Regional Oceanographic Modelling System) model that covered 6 years 2008–2013 (inclusive) (*Chiswell et al., 2019*). Water flows across each cell face cause the movement of nutrients (such as ammonia and nitrate) available to primary producers. The speed and direction of currents influence the spatial distribution of plankton groups. Water temperatures influence biological processes such as respiration (*Hoegh-Guldberg & Bruno, 2010*) (for more details, see Supplemental A). The base TBGB_AM presented here repeated the available ROMS variables as a 6-year cycle. Averaging the ROMS variables across these years was not considered as this could result in implausible physical dynamics, especially with respect to water movement between cells. We ran sensitivities varying the order of ROMS years or repeating one ROMS year to

help understand the effects of inter-annual oceanographic variability on this model, following the methods applied in *McGregor et al. (2019)*.

## Species groups

TBGB_AM uses 51 species groups to model the biological processes. Species groups were defined based on broadly similar form, habitat, and diet. Of these 51 groups, 12 vertebrates, two invertebrates, and one algae comprise single species; all other groups comprised two or more species. The single-species groups are either highly abundant (such as barracouta or seagrass), very distinctive (like fur seals), or for which there are key ecological questions that may require investigation on their own (like snapper and scallops). The main component species of the groups are shown in Tables 1–4. Species group names are intended to be informative but not necessarily restrictive. For example, an invertivores' species group would eat primarily invertebrates, but may also consume a small proportion of vertebrate prey. Species groups modelled with age-structure (all vertebrates and five invertebrate species groups) were modelled using 10 age-classes, with the average expected lifespan of each group split equally into these age-classes. Within each age class, numbers of animals and average weights were tracked. Weights of animal were tracked as mg N, which was split into structural ($S_N$) and reserve ($R_N$) components following the definition in *Broekhuizen et al. (1994)* where reserve weight is the part that can be used during periods of starvation, which includes flesh, fat, reproductive components and other soft tissue. Primary producers and remaining invertebrate groups were modelled as biomass pools (mg N m$^{-3}$) with no age-structure. Weights and biomass-pools were tracked in mg N as nitrogen is the currency used to track the transfer of energy in Atlantis models (*Audzijonyte et al., 2017*). Initial conditions for the species groups were estimated or inferred depending on data and information available. Details on the species groups initial conditions and biological parameters are in Supplemental B.

## Predation

Simulated predation was a four step process which occurred within each cell and at each timestep. From the predator's perspective the steps modelled (as summarised in *McGregor et al. (2019)*) are: (1) Am I allowed to eat it? (2) Is it in the same place at the same time as me? (3) Does it fit in my mouth? (4) How much can I eat? Full details are in the Atlantis User's Guide (*Audzijonyte et al., 2017*). Holling Type II functional feeding response was applied in step 4, which is a function of prey abundance, and predator search rate and handling time.

Diets of each species group were summarised using the same prey categories as in *McGregor et al. (2019)*: Algae, Bacteria, Bird, Cetacea, Coelenterate, Crustacean, Detritus, Echinoderm, Elasmobranch, Microzooplankton, Mollusc, Phytoplankton, Polychaete, Teleost, and Tunicate (Fig. 3). While this summary misses the temporal, spatial, age and size components of the predator-prey interactions, it is useful to check overall diets. For example, warehou (mesopel fish Invert) eat mostly salps (tunicates) as expected; school shark eat mostly fish as expected; flatfish (mostly flounder and sole) eat mostly benthic invertebrates; and invert comm herb (paua and kina) eat mostly algae. Trophic levels were

**Table 1 List of vertebrate species groups for TBGB_AM.** Names in bold at the start of each multispecies group indicates the most dominant species in that group, and it is the species from which productivity parameters for that group were derived. Lifespan is the assumed maximum number of years an individual in that group may live. Ben, benthic; invert, invertivore; lrg, large; mesopel, mesopelagic; pisc, piscivore; sml, small.

| Species group | Main species | Lifespan (years) |
|---|---|---|
| Barracouta | **Barracouta** | 10 |
| Carpet shark | **Carpet shark** | 20 |
| Demersal fish | **Giant stargazer**, Ling, Yellow-eyed mullet, Sea perch, Rattails, Grey mullet, Silver dory, Lookdown dory, Northern bastard cod, Goatfish, Scaly gurnard, Pigfish, Spotted stargazer, Two saddle rattail, Oblique banded rattail, Opalfish, Brown stargazer, Cucumber fish, Swollenhead conger, Giant boarfish, Capro dory, Silverside, Globefish | 20 |
| Elasmobranch invert | **Rough skate**, Skates undefined, Dark ghost shark, Smooth skate, Other sharks and dogfish, Elephant fish, Eagle ray, Stingray, Short-tailed black ray | 10 |
| Elasmobranch pisc | **Thresher shark**, Electric ray, Seal shark, Seven-gilled shark, Bronze whaler shark, Blue shark, Mako shark, Sharks undefined | 20 |
| Flatfish | **Sand flounder**, Greenback flounder, Lemon sole, New Zealand sole, Yellow-belly flounder, Witch, Black flounder, Turbot, Brill, Speckled sole | 5 |
| Kahawai | **Kahawai** | 30 |
| Leatherjacket | **Leatherjacket** | 5 |
| Mackerels | **Jack mackerel (Yellow-tail)**, Jack mackerel (Greenback), Jack mackerel (Peruvian), Blue mackerel | 20 |
| Mesopel fish invert | **Blue warehou**, Silver warehou, Bluenose | 20 |
| Pelagic fish lge | **Trevally**, Albacore, Hoki, Hake, Porcupine fish, Kingfish, Frostfish, Gemfish, Sunfish, Skipjack tuna, Oilfish, Southern boarfish, Ray's bream | 50 |
| Pelagic fish sml | **Pilchard**, Redbait, Anchovy, Garfish, Sprats, Whitebait, Ahuru | 10 |
| Pinniped | **Fur seals** | 20 |
| Red gurnard | **Red gurnard** | 20 |
| Red cod | **Red cod** | 5 |
| Reef fish invert | **Butterfish**, Blue moki, Marblefish, Trumpeter, Banded wrasse, Scarlet wrasse, Wrasse (undefined), Red moki, Copper moki, Seahorse, Spotty, Porae, Long-finned boarfish, Spiny seadragon, Southern bastard cod | 10 |
| Reef fish pisc | **John dory**, Blue cod, Conger eel, Hapuku, Hagfish | 10 |
| Rig | **Rig** | 20 |
| School shark | **School shark** | 50 |
| Seabird | **Seabirds, shorebirds, & black swans** | 20 |
| Snapper | **Snapper** | 50 |
| Southern spiny dogfish | **Southern spiny dogfish** | 30 |
| Tarakihi | **Tarakihi** | 40 |

calculated for each species based on their diets at the prey species group level, summarised over model space, time and species age-classes. The resulting trophic levels ranged from 1 for the primary producers through to 5.49 for elasmobranch piscivores (Fig. 4).

The trophic levels are generally higher than they should be which is due to combining juvenile and adult diets in the trophic level calculation. For example, elasmobranch piscivores predate on juvenile barracouta, but the contribution of barracouta to the trophic level of elasmobranch piscivore uses the trophic level of barracouta averaged over all

**Table 2 List of invertebrate species groups for TBGB_AM.** Description includes main species. Lifespan is the maximum number of years an individual in that group may live. Those groups with no value for lifespan are modelled as biomass pools and hence do not have a lifespan defined as this is only relevant when modelling numbers. Carniv, carivore; Invert comm, commercial invertebrates; herb, herbivore; scav, scavenger; Macrobenth, macrozoobenthos; Meiobenth, meiobenthos; Zoo, zooplankton.

| Species group | Description | Lifespan (years) |
|---|---|---|
| Benthic Carniv | Some gastropod molluscs, polychaetes & crustaceans | |
| Benthic grazer | Benthic animals that consume diatoms and sea grass | |
| Carniv Zoo | Planktonic animals (size 2–20 cm) | |
| Cephalopod | **Arrow squid**, Octopus, Broad squid | 2 |
| Deposit feeder | Detritivores, e.g., some gastropod molluscs, polychaetes, echinoderms (including holothurians) & crustaceans | |
| Dredge oysters | **Dredge oyster** | 4 |
| Filter other | Non-commercial benthic filter feeders, e.g., sponges, bryozoans, ascidians, turbellarians, bivalves, hydroids | |
| Gelat Zoo | Salps, ctenophores, jellyfish | |
| Invert comm Herb | **Paua**, Kina | 6 |
| Invert comm Scav | **Rock lobster**, Paddle crab, Whelks, Sea cucumber | 8 |
| Macrobenth other | Non-commercial benthic organisms (size >1 mm), e.g., polychaetes, echinoderms, sea anemones, crustaceans | |
| Meiobenth | Benthic organisms (size 0.1–1 mm), e.g., nematodes, some small crustaceans | |
| MesoZoo | Planktonic animals (size 0.2–20 mm) | |
| MicroZoo | Heterotrophic plankton (size 20–200 μm) | |
| Mussels | **Greenlip mussel**, Horse mussel | 2 |
| Scallops | **Scallop** | 2 |
| Surf clams | **Cockle**, King clam, Pacific oyster, Pipi, Mactra | 2 |

**Table 3 List of phytoplankton and algae species groups for TBGB_AM.**

| Species group | Description |
|---|---|
| Diatoms | Diatoms (large phytoplankton) |
| Macroalgae | Macroalgae |
| Microphytobenthos | Unicellular benthic algae |
| Pico-phytoplankton | Small phytoplankton |

**Table 4 List of bacteria and detritus species groups for TBGB_AM.**

| Species groups | Description |
|---|---|
| Carrion | Dead and decaying flesh |
| Labile detritus | Organic matter that decomposes at a fast rate |
| Pelagic bacteria | Pelagic bacteria |
| Refractory detritus | Organic matter that decomposes at a slow rate |
| Sediment bacteria | Sediment bacteria |

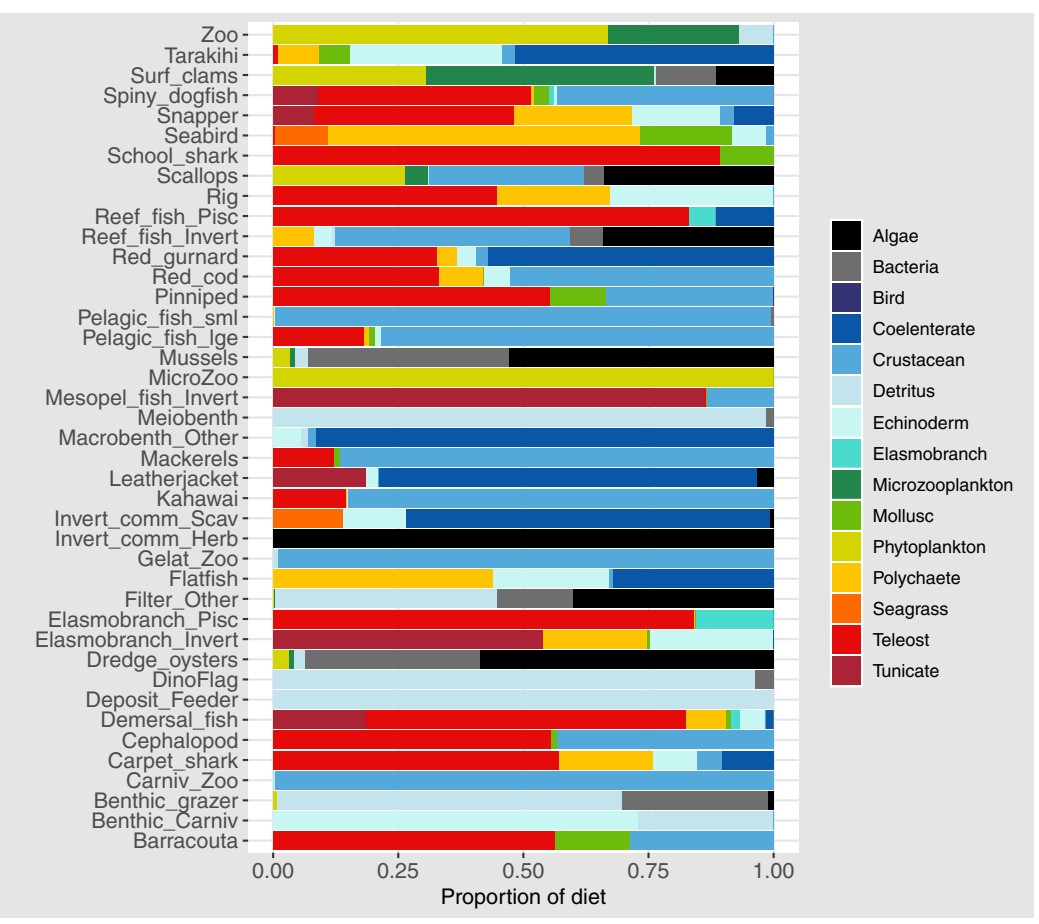

**Figure 3 Summary of the proportion of prey groups in the diets of species groups (Tables 1 and 2) over model years 1900–2014 from the fished model where the proportion is by mg N consumed.**

age-classes, which will be higher than the juvenile trophic level. The effect is confounded throughout the foodweb, as barracouta adults (to continue with this example) also predate on juvenile pelagic fish large, which will artificially increase the trophic level of barracouta.

# ALTERNATIVE MODELS

The alternative models were developed as simplifications of TBGB_AM, although there are additional differences due to the structure of the respective frameworks.

## TBGB_SS

The multi-species size spectrum model was constructed in R (*R Core Team, 2020*). The foundation for the work is the modelling framework *mizer* version 2.0.3, see *Scott, Blanchard & Andersen (2014)*. This is available freely as an R package (*Delius et al., 2020*). For instructions on installing *mizer* and running models, see the *mizer* vignette (*Delius et al., 2020*). The earliest example of using *mizer* to model a multi-species system was by *Blanchard et al. (2014)*, focusing on twelve common pelagic fish species in the North Sea.

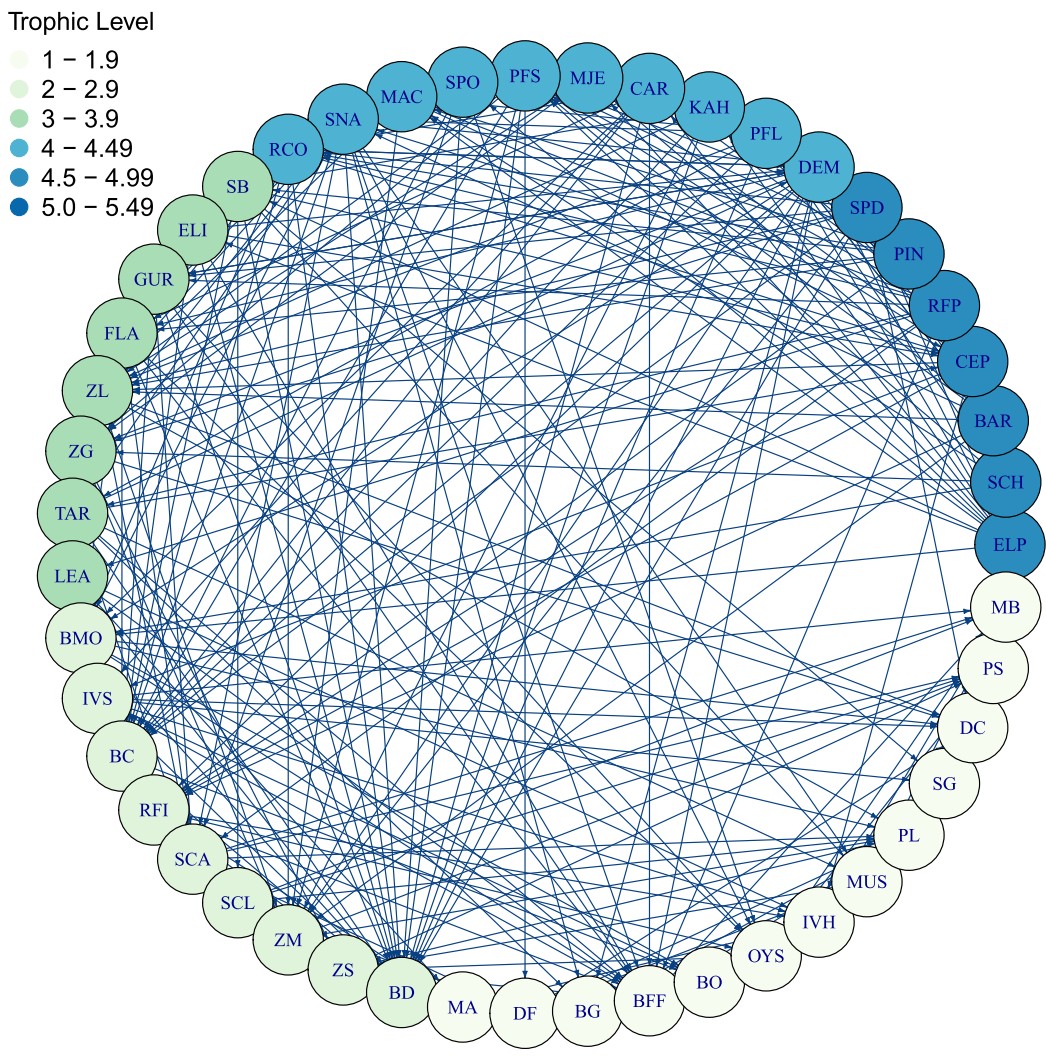

**Figure 4 Foodweb coloured by trophic level from TBGB_AM diets summarised from the model simulation over years 1900–2014, from all model polygons and depth layers, and all species' age-classes.** Species codes (3 or 2 letter) within the shaded circles identify the species functional groups (Tables 1–4). Lines connecting the shaded circles indicate predator/prey interactions. The colour of the shaded circle indicates the trophic level of each species group, with dark blue corresponding to the highest trophic level band (5–5.49), through to light cream corresponding to the lowest trophic level band (1–1.9).

As the first step of setting up the size spectrum model for TBGB, the species list was reduced to those for which a consensus was reached that they fed in a size-based way, and hence it was appropriate to use the size spectrum modelling framework to simulate their life processes. This meant not including the following species in the TBGB_SS model: fur seals, seabirds, benthic carnivores, benthic grazers, carnivorous zooplankton, detritivores, dredge oysters, benthic filter feeders, paua, rock lobster, benthic organisms, planktonic animals, heterotrophic plankton, greenlip mussels, scallops and cockles.

Seven parameters are needed as inputs to the size-structured model for each species. These include:

- the von Bertalanffy growth parameter $k$;
- the allometric length-weight scaling multiplier $a$;
- the allometric length-weight scaling exponent $b$;
- the asymptotic body weight $w_{inf}$;
- the maturity weight $w_{mat}$;
- the preferred predator : prey mass ratio $\beta$;
- the width of the feeding kernel $\sigma$.

The first five of these parameters are available directly from the Atlantis model. For the feeding parameters ($\beta$ and $\sigma$) information was taken from the Atlantis model about the prey species of both adults and juveniles for each species. For the mass of the species doing the feeding, we used asymptotic mass for adults and maturity weight for juveniles. We then approximated the weight of the prey as half of the asymptotic mass for the prey of adults, and half of the maturity weight for the prey of juveniles. We then calculated the predator : prey mass ratio $\beta$ by dividing the former by the latter and taking the average over both juveniles and adults, and calculated the width of the feeding kernel $\sigma$ by taking the $log10$ of the standard deviation of all the mass ratios.

Following this, we calculated the species interaction matrix. We used the spatial overlap of species from the TBGB_AM to estimate the likelihood of interaction between three size combinations of predators and preys (adult predators–adult prey, adult predators–juvenile prey and juvenile predators–juvenile prey), with the juvenile predator–adult prey combination ignored. We then calculated the normalised (0 to 1) overall interaction for each species predator-prey pair using the average of the three predator-prey size combinations.

Using the above parameters and interaction matrix with the software package *mizer*, the following parameters were calculated for each species:

- $h_i(w)$, the maximum intake rate of an individual of species $i$ and weight $w$;
- $\gamma_i$, the search volume for species $i$ at 1g weight;
- $ks_i$, a constant multiplier for metabolic rate.

Also, an extensive description of the default parameter values (which are challenging to calculate empirically) is provided in *Delius et al. (2020)*. As part of the model set up, the carrying capacity of the resource spectrum was reduced to 70% of that of the North Sea model (*Blanchard et al., 2014*), as it was observed that feeding levels (even for large individuals) in the TBGB model were too dependent upon the resource rather than other species.

The final step for the model was to tune free parameters related to reproduction ($e_{repro}$ and $R_{max}$) to fit the unfished (virgin) biomasses of species to those generated in the Atlantis model. The decision was taken to fix $e_{repro}$ to 1 for all species, and modify $R_{max}$ for
each species until a good fit was observed for all species biomasses. Pearson's coefficient was used to measure closeness of fit between model biomasses and those of the Atlantis model.

## TBGB_EwE

TBGB_EwE was developed in two parts, a mass balanced model (Ecopath) and a simulation model (Ecosim). TBGB_EwE included the same 51 species functional groups used in TBGB_AM and were modelled as biomass pools (in t/km$^2$). Diet and prey preferences, fishing mortality and initial conditions were all based on those used in TBGB_AM.

TBGB_EwE used a simpler spatial structure than that used for TBGB_AM, where all functional groups were assumed to inhabit the entire study area. The vulnerability of prey groups to predation and predatory searching rates was used as a proxy for the variability of preys and the different levels of interaction between species (predators and prey). Diet was also used as a proxy for spatial dynamics in terms of migration, where a proportion of the total diet could be apportioned to being outside the study area.

Realised diets in EwE rely on the assumption that what is consumption for one group is mortality for another. Ecotrophic efficiency is used as indication of how heavily a group is being preyed upon and whether enough individuals are available to die of old age. Diet must be input as the proportion of each prey group in a predator's diet. These proportions can be varied to allow for successful mass balance but builds basis of consumption rates and system dynamics. The relative proportion of prey groups in predator diets was different to those used in TBGB_AM, but since the dietary components remained the same, realised diets were comparable. The single biggest difference between the two diets related to modelling bacterial groups which affected the diets of those groups that consume bacteria.

Although it is possible to model age structure in EwE, this functionality was not used for TBGB_EwE, since interactions between adults and juveniles are not well simulated. For example, if there is less food for juveniles or more mortality (both resulting in less juvenile biomass) the adult population remains unaffected.

## Comparison of alternative models

Table 5 presents key aspects of the alternative models as they compare to TBGB_AM. Neither of the alternative models developed have space explicitly defined. They do, however, both use an availability term which acts as a proxy for some spatial dynamics such as providing refuge from predators or fishing that might be expected from spatial separation. TBGB_SS sets an availability term for each pair of species groups that defines how much they are expected to spatially overlap. These were estimated using the spatial overlap from the base TBGB_AM, averaged over the model years 1900–2014. These availabilities were not age- or size- structured, whereas the spatial distributions are in TBGB_AM.

Species groups were another key difference. TBGB_EwE modelled the same 51 species groups used in TBGB_AM, although as biomass pools (in t/km$^2$) with no age structure.

**Table 5  Overview comparison of model structure for TBGB_AM (Atlantis model), TBGB_EwE (Ecopath with ecosim model) and TBGB_SS (size spectrum model).**

| | TBGB_AM | TBGB_EwE | TBGB_SS |
|---|---|---|---|
| Species | 51 groups; age-structured; individuals and biomass pools | 51 groups; no age-structure; biomass pools | 22 groups + primary producer; size-structure; individuals |
| Spatial | 25 dynamic polygons, 5 water column depth bins | Not spatial, availability term acts as proxy | Not spatial, availability term acts as proxy |
| Temporal | 12-hour time steps; adaptive time steps for high rate-of-change components such as nutrients; burn-in 1865–1899; model hindcast 1900– 2014 | Annual timesteps; model hindcast 1959– 2014 | 10 time steps per year; model hindcast; 1900–2014 |
| Predation | Prey availability matrix; spatial/ temporal overlap; gape size; feeding dynamics (handling time, searching time, maximum growth, prey abundance); assimilation efficiency; habitat refuge | Prey availability matrix; habitat refuge | Prey availability matrix; size-based (similar to gape size) |
| Environmental | Forcing of sea temperature, salinity and flux; potential to force light, sediments, nutrients, and habitat | Potential to force as time-series e.g. changes in sea temperature or primary production | Potential to force changes in primary production |
| Fishing | Currently forced removals, defined by fleet, month, polygon, species functional group, and age-class; potential for effort-based; potential to link economic component | Forces using a time-series of fishing mortality F; values from TBGB_AM outputs; defined by fleet and species group | Forces using a time-series of fishing mortality F; values from TBGB_AM outputs; defined by species group |

TBGB_SS modelled all 21 of the vertebrate species groups defined for TBGB_AM, as well as cephalopods. For these groups, the size-structured nature of this framework allows for variation in growth rates, diet and predation vulnerability throughout each groups' life span, from egg to adult. However, the size-structured model does not explicitly define or capture the dynamics of benthic invertebrate functional groups.

Historical catches were forced in all three models, but in slightly different ways due to the structure of the models. TBGB_AM removed numbers of fish to match historical tonnes caught, and these were calculated within TBGB_AM based on the weights of individuals at each age class, with the total catch removed apportioned to age-classes based on selectivities defined by age-class. TBGB_EwE removed catches as biomass, and there was no age-structure, so no selectivities with respect to age or size were required. TBGB_SS removed catches using an F time-series for each species group, which was based on estimates from TBGB_AM. TBGB_SS had knife-edge selectivity such that fish were only caught at or above the size at maturity.

During model calibration, each of the models focused on different aspects of the model. TBGB_SS focused mostly on adjusting reproductive outputs for each species, and the planktonic primary production to achieve virgin biomasses matching those in the TBGB_AM model. TBGB_AM focused mostly on prey availabilities, and also additional

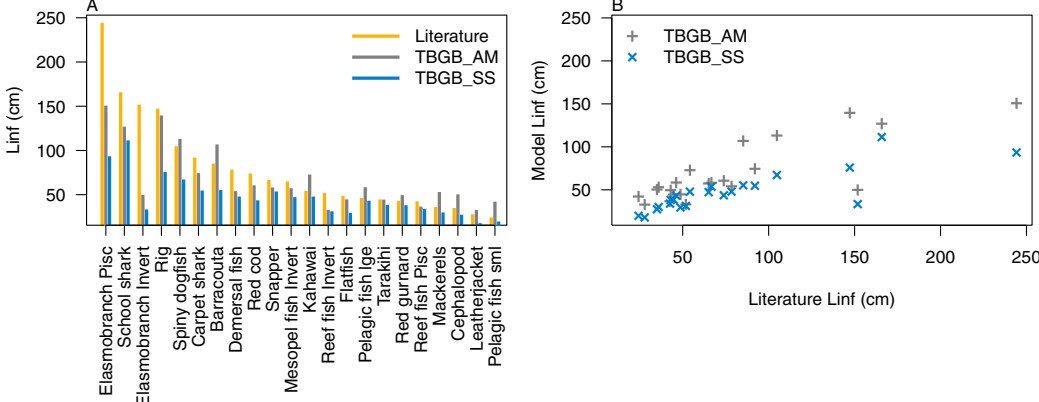

**Figure 5** $L_\infty$ from the literature (orange bars), TBGB_AM (grey bars) and TBGB_SS (blue bars) (A), and TBGB_AM $L_\infty$ plotted against literature $L_\infty$ (grey pluses) and TBGB_SS $L_\infty$ plotted against literature $L_\infty$ (blue crosses) (B).

(non predation) mortality for higher trophic levels. TBGB_EwE focused on balancing energy inputs (production) with outputs (consumption).

## CALIBRATION

Calibration of TBGB_AM included ensuring stable biomass trajectories when applying no fishing; realistic realised diets; realistic growth and mortality (size-at-age and proportions-at-age), following the methods and recommendations of *Pethybridge et al. (2019)*, and *McGregor et al. (2019)*. Parameters tuned during calibration are listed in Supplemental J, and generally consisted of growth, mortality and predation parameters.

Biomass trajectories should reach a quasi-equilibrium when modelled with constant oceanography and no fishing (*Kaplan & Marshall, 2016*). While oceanography is not constant in our non-fishing model as it changes by year (Oceanography), most of the age-structured groups should maintain their dynamic stability. For the TBGB_AM base model, all biomass trajectories remained within 20% of their coefficients of variation (CVs) over the simulated 1900–2014 model period. Biomass trajectories for all age-structured groups from the un-fished model are in Supplemental C.

Atlantis simulates growth rates of age-structured groups as a function of consumption. If growth is too slow, there may be insufficient food available, the feeding search rate could be too low or handling time too high, and the reverse of these when growth is too fast. In the TBGB_SS model growth is also a function of consumed prey, taking into account metabolism and movement (*Scott, Blanchard & Andersen, 2018*). Allocation of consumed prey to reproduction is set such that growth approximates the von Bertalanffy curve at a constant feeding level (*Hartvig, Andersen & Beyer, 2011*). Simulated growth rates of age-structured species groups were assessed by comparing the simulated size-at-age with those expected based on growth curve estimates from the literature (Supplemental B), and those resulting from the size-spectrum model. The full growth curves are in Supplemental D, and summary figures of the maximum expected size for each species functional group ($L_\infty$) are in Fig. 5. We used the upper 90th percentile for weight,

converted to length using the length-to-weight conversion parameters (Supplemental B), from TBGB_AM outputs from 1900–2014 as $L_\infty$, and the maximum size from the base un-fished model at equilibrium for TBGB_SS. While many were in line with the literature, both models generally produced smaller maximum sizes than what the literature suggests for the larger species such as elasmobranch piscivores and school sharks, and TBGB_AM generally produced larger than expected maximum sizes for smaller species groups such as pelagic fish small, cephalopods and mackerels.

In a closed system, natural mortality should ideally be entirely intrinsic within the model (e.g. from predation, starvation, and light, oxygen or nutrient deprivation). As this was not the case in TBGB_AM, additional mortality was applied for some species groups to allow for natural mortality suffered either outside of the model domain, or not captured by the model dynamics. Following the approach of *McGregor et al. (2019)*, age-structured simulated natural mortality rates from the stable base model were compared to estimates of $M$ from the literature where available (Supplemental B) by comparing the proportions-at-age with the corresponding exponential decay curve. The overlaid simulated and 'observed' were generally very similar (Supplemental E), although kahawai stood out as having less natural mortality in the model than the literature would suggest, as did (although to a lesser extent) mesopelagic fish invertivores.

## SENSITIVITY ANALYSES

A sensitivity analysis of the TBGB_AM was carried out to assess the uncertainty of the model inputs, the propagation of this uncertainty to the outputs and therefore its effects on model performance. This analysis was performed on the following components of the TBGB_AM

### Initial conditions

Initial conditions were perturbed for TBGB_AM following the methods of *McGregor, Fulton & Dunn (2020)*. The resulting simulations were then used for assessing responses of TBGB_AM to historical fishing, thus allowing us to present these results as envelopes of plausibility rather than single trajectories. The species groups with the largest between-run CVs by the end of the simulation period (2000–2014) were generally lower in the foodweb, such as zooplankton, deposit feeders and picophytobenthos (Fig. 6). In addition to these, scallops and flatfish also featured among those with high CVs, and for both of these it was due to an apparently fine line between the stock crashing under fishing pressure, or persisting (Fig. 7).

### Oceanographic variability

The effect of recycling oceanographic variables was assessed following the methods of *McGregor et al. (2019)*, which has two parts: (1) establishing confidence intervals for our model simulations with respect to oceanographic variability; (2) assessing the effect of repeating oceanographic variables from any one year, and whether these take the model outside of the established confidence intervals.

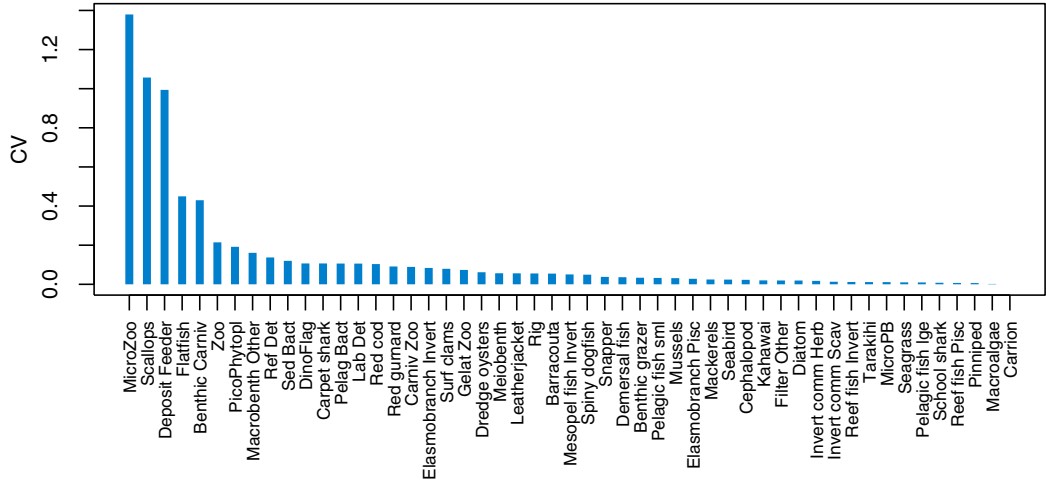

Figure 6 Between-run CVs from 2000–2014 that resulted from perturbing the initial conditions of TBGB_AM.                                   

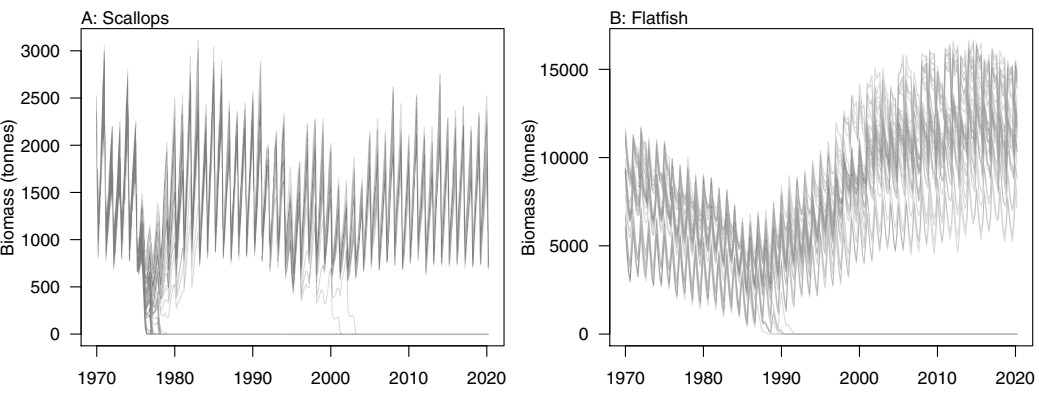

Figure 7 Simulated biomass trajectories from TBGB_AM under perturbed initial conditions for (A) scallops and (B) flatfish.                   

Part one sampled ROMS years at random with replacement for each model year simulated (bootstrapped the ROMS years) and repeated this for 50 model runs; part two repeated one ROMS year for all model years simulated and did a separate model run for each of the six ROMS years. In both cases, the 2008 ROMS was repeated for a 35-year burn-in period, followed by a 115 year simulation. The 2008 ROMS year was chosen for the burn-in period as this year seemed to be representative of the means from all ROMS years for sea temperature and salinity when averaged over the model area. The full set of figures with temperature and salinity from each ROMS year, overlaid with averages from all ROMS years are in Supplemental A. Figure 8 shows the sum-of-squares between the values for each ROMS year and the average for salinity and temperature as well as the Pearson's correlations. The 2008 ROMS year had the lowest combined sum-of-squares, although correlation for salinity was not as high as the other years. Bootstrapping the ROMS years was used to establish confidence intervals with respect to between-year oceanographic variability. Repeating each ROMS year in turn was testing the effect of

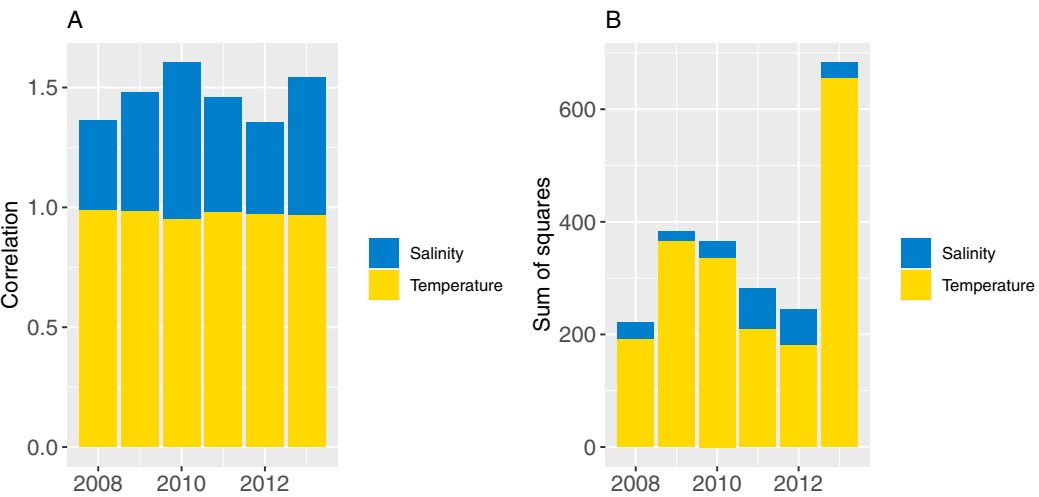

**Figure 8 Correlation (A) and sum-of-squares (B) for salinity (blue) and sea temperature (yellow) variables for each ROMS year with respect to the relative averages from all ROMS years.**

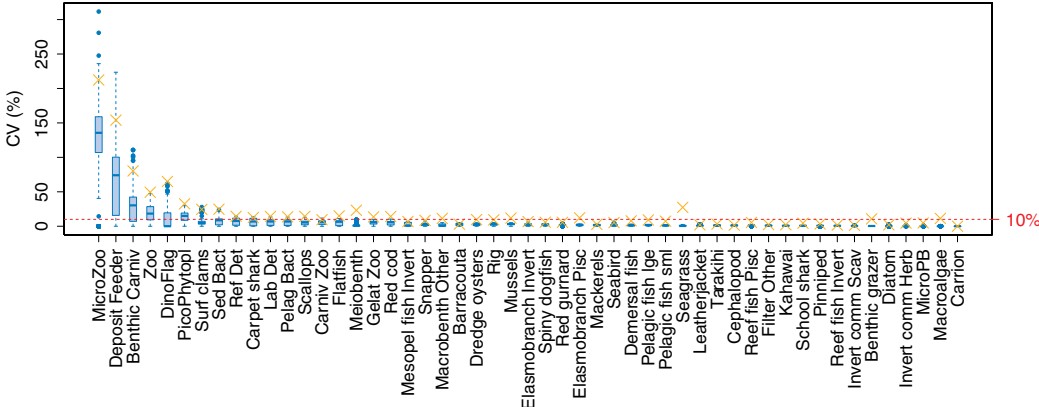

**Figure 9 Between-run CVs for biomass output trajectories for each species group in TBGB_AM, from 50 model runs with ROMS years (2008–2013) sampled at random with replacement for each model year 1900–2014 (blue boxes) and upper 90% CIs for between-run CVs from repeating each ROMS year for the 1900–2014 model years (gold crosses).**

multiple years being different to the other years in some consistent way, such as cooler or warmer.

The established biomass confidence intervals established from varying the oceanographic variables were fairly narrow for most species groups, with CVs < 10% (Fig. 9). Of the exceptions, micro-zooplankton had the highest CVs extending up to almost 250%, followed by deposit feeders, benthic carnivores, meso-zooplankto, dinoflagelettes, and pico-phytoplanton. That these groups were found to be most sensitive to oceanographic variability in the model is a plausible and sensible result. When we compared the between-run CVs, we found these varied more than when sampling ROMS years randomly (Fig. 9), suggesting strong effects from repeating the ROMS variables from

any one year. Dinoflagelettes, seagrass and meiobenthos all had greater between-run CVs from repeating ROMS years than from sampling ROMS years.

## Connectivity and influence

Understanding which species groups are most influential in the model is another test for realistic dynamics, and is another way we can compare the dynamics of the models. EwE models output several measures of keystoneness, with $KS_3$ recommended to be 'best' by *Valls, Coll & Christensen (2015)* following analyses of many variations on keystone ranking calculations. For TBGB_AM, we followed the simulation method applied in *McGregor et al. (2019)*, but modified the keystoneness calculation to match that recommended in *Valls, Coll & Christensen (2015)* and applied in EwE (Eq. 1). For the TBGB_AM simulations, we perturbed each species group in turn by adding additional natural mortality, then analysed the responses of the other groups in the system after a 50-year simulation period. Two levels of additional mortality were applied: $M$(per year) + (0.1, 0.005), and the resulting biomasses analysed with respect to the Base Model to calculate keystone rankings, which were then compared to those produced for TBGB_EwE.

$$KS_i = log \left( \sqrt{\sum_{i \neq j} m_{ij}^2} \times drank_i \right) \qquad (1)$$

$m_{ij}$ is the impact of species $i$ on species $j$,
$drank_i$ is the rank of species $i$ in descending order or biomass.

The top species for keystoneness are not the same for TBGB_AM which has mussels, demersal fish, seabirds, and dredge oysters, and TBGB_EwE which has reef fish piscivores, elasmobranch piscivores, pelagic fish large and snapper (Fig. 10). The Pearson's correlation between the two sets of rankings is 0.19, suggesting a weak and likely insignificant correlation. If we lower the benchmark to within the same third for keystone ranking (e.g. top, middle, or bottom third for both models), 11 out of the 30 age-structured species groups are within the same third; and seven are in the opposite third.

## FISHING

Catch histories from 1900 to 2013 were estimated for all of the species groups that have been commercially exploited. Catches required partitioning to provide catch by species group, by month, by fleet, and by cell. The available catch data seldom provided this level of detail, so numerous assumptions were necessary to develop catch histories (more details in Supplemental Materials). The model operated on a one-day cycle, so catches were actually required for this time interval. However, it was considered that an estimation of catches by month (subsequently split into daily amounts) would be sufficient to describe the patterns of seasonal variation in commercial catch apparent for most of the species groups.

Catch histories for the commercially exploited bivalve species (scallops, oysters, mussels and surf clams) were developed using the units tonnes meatweight (rather than greenweight (unshucked shellfish)). Meatweight landings of scallops have been recorded, and the derivation of the catch history for this group is given below. Landings of the

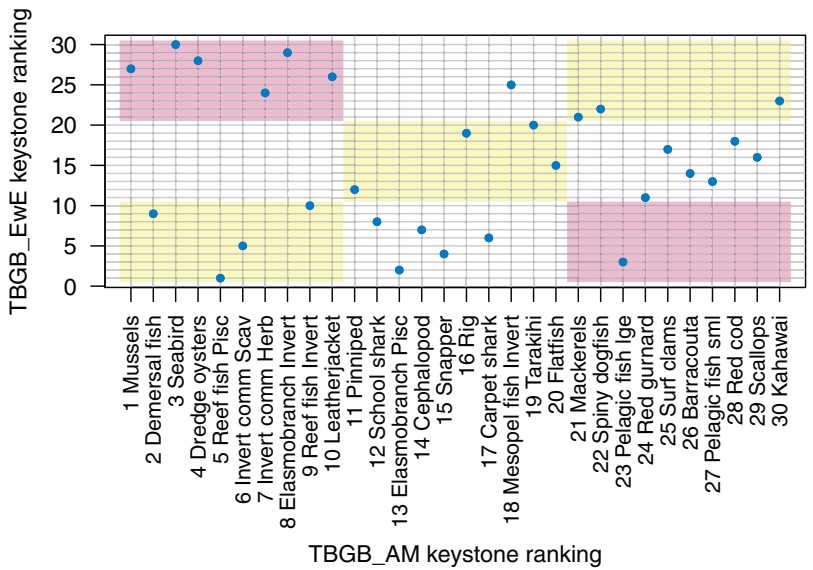

**Figure 10 Keystone ranking from TBGB_AM (*x*-axis) and TBGB_EwE (*y*-axis) for all age-structured species groups, with numbers giving keystoneness ranking (1 is the most influential using Eq. 1).** Shading indicates species groups that are at least within the same third of keystone ranking in both models (e.g. top third, middle third, bottom third) (yellow), and those that are in opposite thirds (e.g. top third in TBGB_AM and bottom third in TBGB_EwE or top third in TBGB_EwE and bottom third in TBGB_AM) (red).

other three bivalve groups are recorded as greenweight. Catch histories for these were derived (in tonnes) for polygon and month as described above for wetfish from the FSU and QMS databases. The greenweights were then adjusted to meatweight using the following conversion factors: dredge oysters, 0.12; mussels, 0.25; surf clams, 0.18.

The commercial catch history for scallops was developed using data from *Williams et al. (2014)*. Data from 1978 onwards were reported by scallop reporting sector (see Fig. 3 of *Williams et al. (2014)*), and these were allocated to polygon and to month. Catch was allocated amongst months in each polygon using the proportions used to derive Fig. 10 of *Williams et al. (2014)*. Catches from 1959 to 1977 were not available by area, so were allocated to polygons based on the mean distribution of catches after 1977.

Reliable estimates of recreational catch from the TBGB region are sparse; *Cole et al. (2006)* estimated shellfish harvest in 2003–2004, *Davey et al. (2008)* estimated harvest of snapper and blue cod in 2005–2006, and a National Research Bureau survey estimated catches of all species in 2011–2012 (B. Hartill, NIWA, 2017, personal communication). Other estimates are available (see *Ministry for Primary Industries (2017)*), but they are generally not considered reliable (B. Hartill, NIWA, 2017, personal communication). It is acknowledged that the recreational catch in this area is very dynamic, with factors like weather and localised abundance of species driving harvest levels.

## SKILL ASSESSMENT

A research trawl survey series conducted in 11 years between 1992 and 2013 has sampled in Tasman and Golden Bays (*Stevenson & Hanchet, 2000*; *Stevens et al., 2017*). Biomass

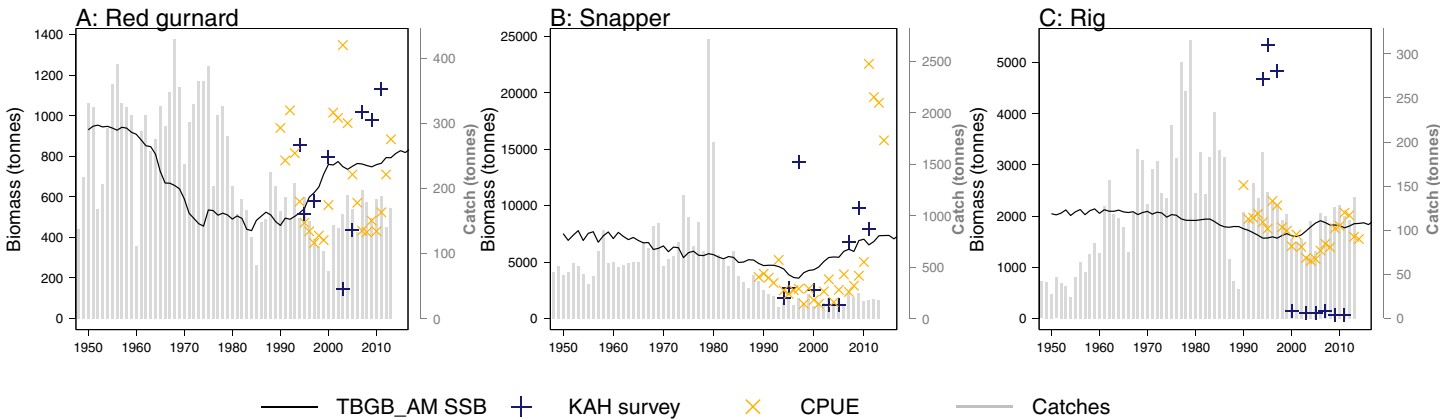

**Figure 11 TBGB_AM estimated spawning stock biomass (SSB) (black solid), survey estimated biomass (midnight blue pluses), and CPUE (orange crosses) where available for the red gurnard (A), snapper (B), and rig (C).** CPUE and survey biomass estimates were rescaled to match the mean of the TBGB_AM estimated SSB. Grey bars are estimated catches that were forced removals in TBGB_AM.

estimates from strata 17, 18, and 19 (approximating Statistical Area 038 which is the TBGB_AM model area) were compared to the biomass outputs from TBGB_AM. Three of the species groups (red gurnard (*Starr & Kendrick, 2017a*), snapper (*Langley, 2018*) and rig (*Starr & Kendrick, 2017b*)) have CPUE (catch per unit effort) that we have also compared to the corresponding TBGB_AM biomass outputs. There is a dredge survey for scallops in Tasman and Golden Bays, conducted annually in May–June (*Williams et al., 2014, 2015*) for which the dredge efficiency was revised in *Tuck, Williams & Bian (2018)*. It is the revised biomass index we have used to compare with the TBGB_AM scallop biomass. The full set of figures with survey biomass estimates overlaid on TBGB_AM biomass outputs are in Supplemental G. The three species groups with corresponding CPUE and survey biomass estimates are presented in Fig. 11. As a quantitative measure of comparison, we calculated Pearson's correlations (Table 6). We used Pearson's which is one of the metrics used for other Atlantis model skill assessments (*McGregor et al., 2019*; *Sturludottir et al., 2018*; *Olsen et al., 2016*). The correlations between the model and survey range from 79% for school shark, down to −79% for rig. Rig were also negatively correlated with the CPUE index at −37% (Table 6). The rig survey biomass suggests the population was very low from 2000–2011, whereas TBGB_AM was stable with a slight increase in biomass for these years. The CPUE for rig also suggested a decline in the early 2000s, followed by an increase which is not apparent in the survey or TBGB_AM. The other negatively correlated species were cephalopods (−15%), and tarakihi (−20%), neither of which had an apparent trend in the survey biomass or the TBGB_AM biomass. Nine out of the 19 speices groups with survey biomass estimates had positive correlations of greater than 20%. Only snapper was positively correlated with CPUE (66%).

Table 6 **Pearson's correlation between TBGB_AM spawning stock biomass (SSB) and survey biomass estimates (Survey) and fishery catch per unit effort (CPUE) (rounded to two significant figures).** Invert, invertivore; mesopel, mesopelagic; pisc, piscivore.

| Species group | Survey | CPUE |
|---|---|---|
| Barracouta | −0.05 | NA |
| Carpet shark | 0.43 | NA |
| Cephalopod | −0.15 | NA |
| Demersal fish | 0.48 | NA |
| Elasmobranch Invert | 0.16 | NA |
| Elasmobranch Pisc | 0.43 | NA |
| Flatfish | 0.3 | NA |
| Leatherjacket | 0.061 | NA |
| Mackerels | −0.067 | NA |
| Mesopel fish Invert | 0.55 | NA |
| Red cod | 0.41 | NA |
| Red gurnard | 0.15 | −0.059 |
| Reef fish Pisc | −0.084 | NA |
| Rig | −0.79 | −0.37 |
| Scallops | 0.36 | NA |
| School shark | 0.79 | NA |
| Snapper | 0.13 | 0.66 |
| Spiny dogfish | 0.27 | NA |
| Tarakihi | −0.2 | NA |

# BRINGING IT TOGETHER: COMPARING MODELLED ECOSYSTEM DYNAMICS

## Realised diets

We compared realised diets from the three models, using the base un-fished version, at equilibrium for TBGB_SS, and summarised over 1900–2014 and 1959–2014 for TBGB_AM and TBGB_EwE respectively. The full set of figures showing proportion of weight consumed by prey species group for each of the three models are in Supplemental H. We dropped plankton diet proportions for TBGB_SS as they swamped the diets due to large amounts of this being eaten when the animals are very small, as is the nature of a size-spectrum model. Having done this, the next smallest prey group (pelagic fish small) then dominated all diets. They were even the largest prey group for predators that don't generally consume small pelagic fish, such as reef fish invertivores, elasmobranch invertivores, carpet sharks and flatfish. As a result, the diets can only be similar between the size-structured model and the other two models for predators that eat a lot of small pelagic fish. The realised diets from TBGB_EwE don't seem to show any changes from the input diets, so these do not reflect anything of the model dynamics, but rather how the diets were specified for the model. The TBGB_AM realised diets are the most complex, and achieving realistic realised diets was one of the goals of model calibration. The results

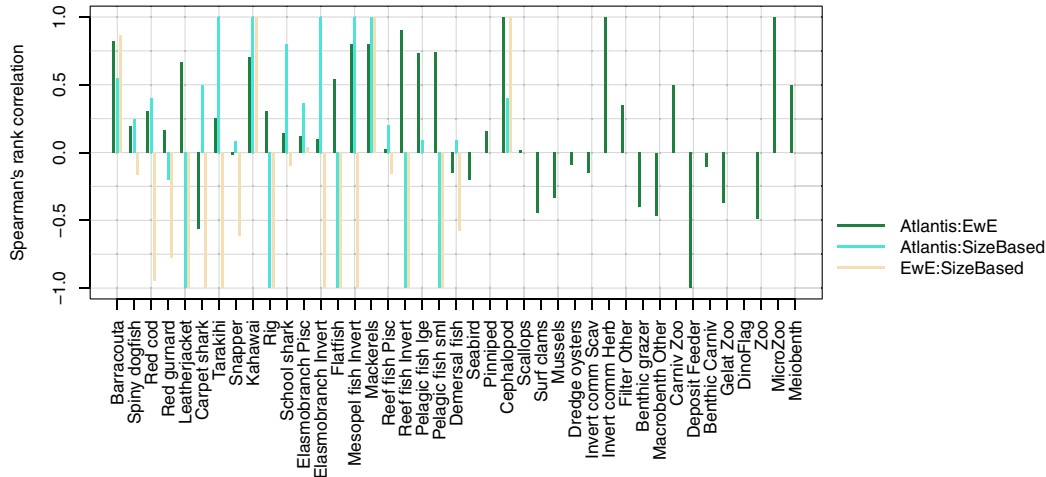

**Figure 12 Spearman's rank correlation comparing realised diets from TBGB_AM with TBGB_EwE (dark green bars); TBGB_AM with TBGB_SS (aqua bars) and TBGB_EwE with TBGB_SS (beige bars).** Diets with zero overlap were assigned a value of −1.

presented for the model comparison have been summarised over model space, time and species age, which loses a lot of the detail, but is a neccesary abstraction for the purpose of comparison. We calculated Spearman's rank correlation of the diets as direction and monoticity between diets seemed more appropriate than to test for a linear relationship. We assigned a value of −1 to any compared diets with zero overlap. There were five species groups where the size-structured model had zero overlap with the other two models due to these predators only eating pelagic fish small in the size-structured model, and not eating pelagic fish small' in the other two models. These were leatherjacket, rig, flatfish, reef fish invertivores and pelagic fish small. Between the Atlantis and EwE models, the invertebrates were more likely to have negatively correlated diets, with the exception being carpet shark, found to be eating more red cod in TBGB_AM and less demersal fish and invert commercial scavengers than in TBGB_EwE (Fig. 12).

## Trophic level

We compared trophic levels calculated from realised diets for TBGB_AM and TBG_EwE. We did not calculate trophic levels for TBGB_SS due to the differences in diets, the limited number of species groups modelled, and the focus of the size-structured model on animals progressing through the trophic levels as they grow. While the latter is somewhat applicable to the Atlantis model as prey preferences are separately defined for juveniles and adults, and spatial, temporal, habitat and gape sizes allow for further differences in diet between age-classes, the coarse scale of up to 10 age-classes for each species group makes the realised diets and hence trophic levels more comparable with the EwE model. Nonetheless, there were some systemic differences in trophic level between TBGB_AM and TBGB_EwE, such as the higher trophic levels generally presenting with inflated trophic levels in the Atlantis model (Fig. 13). For example, elasmobranch piscivores have trophic level 5.5 in TBGB_AM and 4.8 in TBGB_EwE. The difference seems to be largely due to the fairly high presence of macrobenth other in the TBGB_EwE diet that is not

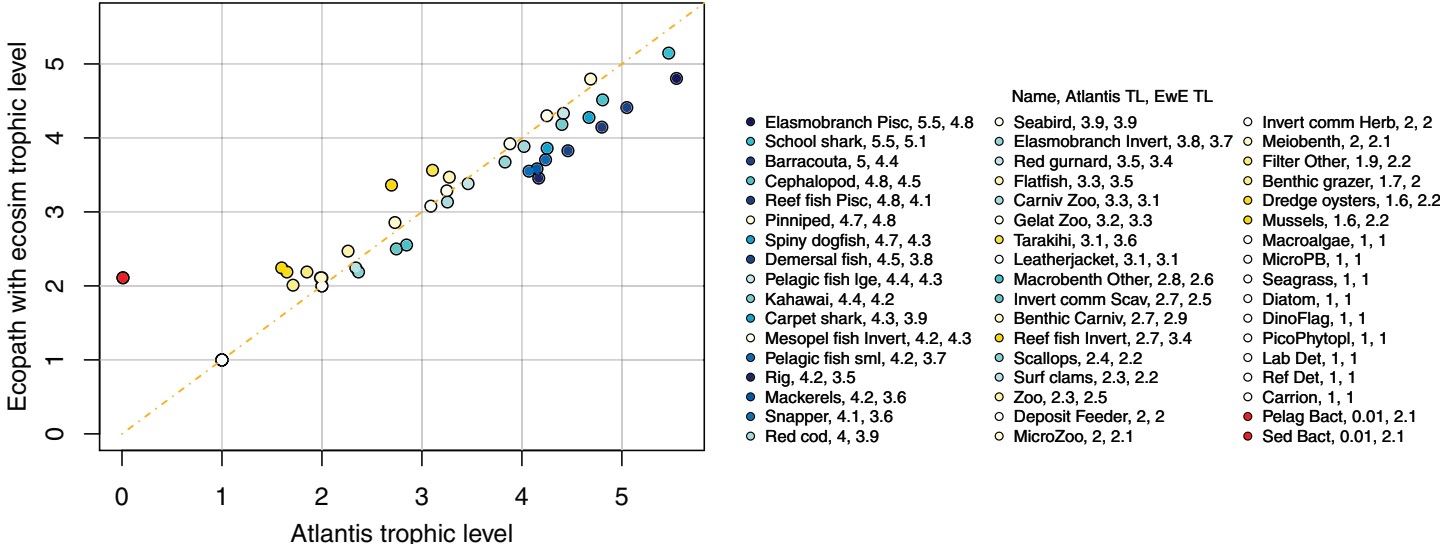

**Figure 13 Trophic levels (TL) calculated from TBGB_AM (*x*-axis) and from TBGB_EwE (*y*-axis).**

apparent in TBGB_AM. School shark also have a higher trophic level in TBGB_AM at 5.5 compared to 5.1 in TBGB_EwE, and this is likely due to larger proportions of cephalopods and gelatenous zooplankton in the TBGB_EwE diet, and the higher proportions of barracouta and mackerels in the TBGB_AM school shark diet. There is also a difference for bacteria as it was categorised as a predator in TBGB_EwE, but not in TBGB_AM in which we have given it a nominal close-to-zero trophic level of 0.01. Bacteria are consumed in TBGB_AM, but not in TBGB_EwE, so the bacteria trophic level affected other trophic levels in TBGB_AM but not in TBGB_EwE.

## Responses to fishing

We forced historical fishing in all three models, then compared the resulting biomass trajectories. The overlaid figures are in Supplemental I, and we have summarised the comparisons using Pearson's correlation (Fig. 14). Some of the species groups have very high (>70%) correlation between all three models. These are elasmobranch invertivores, mackerels, mesopelagic fish invertivores, schools sharks, and spiny dogfish. There were no strong negative correlations between TBGB_EwE and TBGB_SS, but seven of the species groups had negative correlations of more than 50% between TBGB_AM and either TBGB_EwE or TBGB_SS. These were barracouta, carpet shark, elasmobranch piscivores, red gurnard, invert comm herbivores, leatherjacket and tarakihi. The ways in which they were different varied. For example, barracouta decreased under fishing for both TBGB_EwE and TBGB_SS, but gave no response for TBGB_AM, which could be due to migration in the Atlantis model buffering the effects of fishing. Carpet shark had fluctuations in biomass from 1980 in TBGB_SS that don't correspond to the time of direct fishing, and were not present for TBGB_AM or TBGB_EwE. There was a similar situation for elasmobranch piscivores, with more variation with respect to time in TBGB_SS than would be expected as direct fishing effects, and that were not present in TBGB_AM or

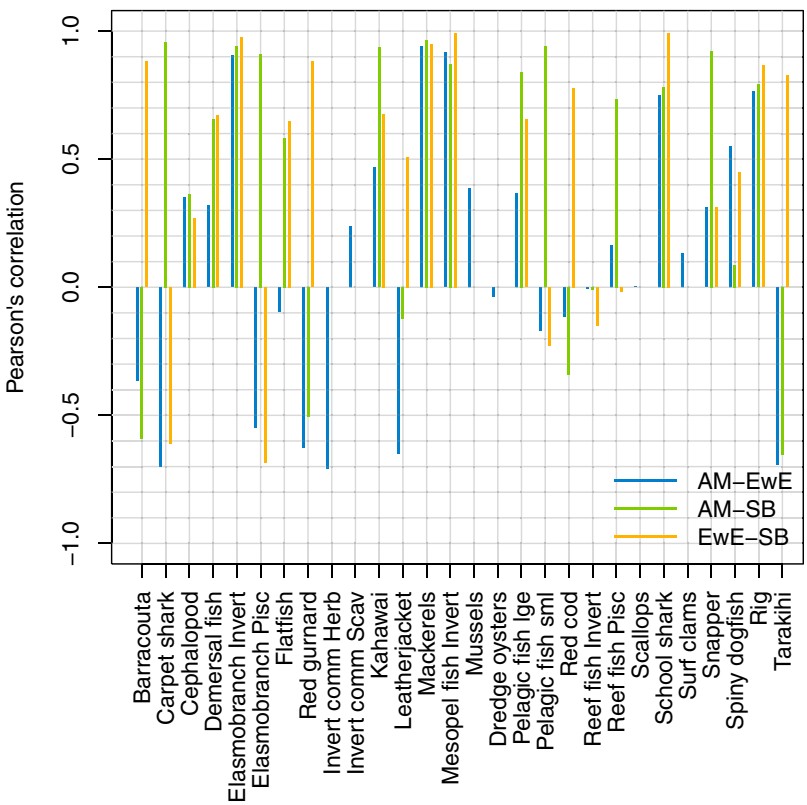

**Figure 14 Pearson's correlation between model simulations in response to forced historical fishing.**

TBGB_EwE. Red gurnard increased during the period of heaviest fishing in TBGB_AM, then declined under lighter fishing that followed, whereas the responses in TBGB_AM and TBGB_SS were more indicative of direct fishing effects as the biomass declined under heavy fishing then recovered under light fishing. Invert comm herbivores decreased briefly in years with higher catches in TBGB_AM, with quick recoveries, but gave no response to fishing in TBGB_EwE, and were not included in TBGB_SS. Leatherjacket had a slightly decreasing biomass trend in TBGB_SS and TBGB_EwE, but remained flat for TBGB_AM. Tarakihi declined under the heaviest fishing in TBGB_SS and TBGB_EwE and recovered under lighter fishing, while in TBGB_AM the biomass signal was almost opposite.

## Diversity

We calculated the modified version of Kempton's Q diversity index, as available in EwE (*Christensen, 2009*) to all three models. The response to historical fishing at the diversity level was very similar between TBGBG_AM and TBGB_EwE, but different in TBGB_SS (Fig. 15). All three models signal a decline in diversity under fishing, but whereas there is only a decline in TBGB_SS, both TBGB_AM and TBGB_EwE show an initial increase in diversity under fishing which was then followed by a decline from the mid-late 1980s.

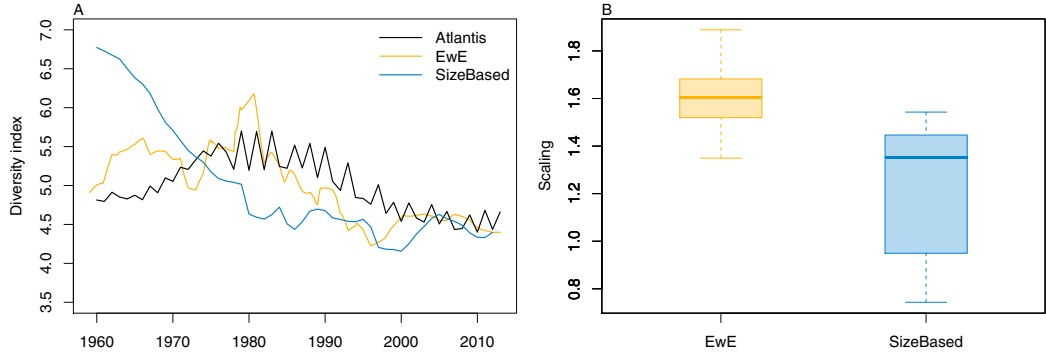

**Figure 15 Diversity index for TBGB_AM (black line), TBGB_EwE (gold line) and TBGB_SS (blue line), scaled to the same mean (A) and scaling ranges given by TBGB_EwE diversity/TBGB_AM diversity (gold) and TBGB_SS diversity/TBGB_AM diversity (blue) (B).**

## DISCUSSION

Key to this work, is to what degree the three developed ecosystem models of the TBGB ecosystem agree, and to what end they might be useful. Both of the terms 'agree' and 'useful' are open to interpretation. Testing for agreement between the models has been carried out in this study with respect to dynamics at the species group level, such as diets and growth rates, and with respect to dynamics at the system level such as trophic level and diversity. We will discuss model usefulness based on limitations due to model structure, intended purpose of the respective modelling frameworks, and concerns that have arisen through model validation and comparison carried out here.

The ecosystem models developed in this work are all useful tools for helping us learn about and understand the TBGB ecosystem. We can use these models to explore scenarios designed to understand the flow-on effects of different pressures on the ecosystem. As the models vary in structure and complexity, they are not equally suited for all scenarios (Table 7). Where more than one model is suited to explore a scenario, these models may highlight different aspects of the ecosystem's response due to differences in structure and how these follow through to affect dynamics. When presented with different outcomes, the question should not be which model is more 'right', but rather what can we learn from the outcomes of each model, and how do they relate to each other. Using multiple models can give us a more extensive understanding of the ecosystem responses, hence it is ideal to use more than one model to explore a scenario where it is possible to do so. Proxies can be used to explore scenarios in TBGB_AM and TBGB_SS that they are not directly suited for. For example, coastal erosion could be explored through reduced habitat, and increased sedimentation could use additional mortality for animals that are expected to be vulnerable to this change.

The alternative models used parameter values from the Atlantis model, such that comparisons between the three models would relate to model structure rather than parameterisation. For some parameters these were inputs to the Atlantis model from available data or the literature, e.g. lifespan or age at maturity, and these should be the same if the alternative models were developed independently to the Atlantis model. Some

**Table 7 Overview of model use suitability for TBGB_AM (Atlantis model), TBGB_EwE (Ecopath with ecosim model) and TBGB_SS (size spectrum model).** ✓: the model is suited for the scenario; ✗: the model is not directly suited for the scenario, although in some cases a proxy may be used. An extension to the mizer package used for TBGB_SS has been developed to include temperature effects (*Woodworth-Jefcoats, Blanchard & Drazen, 2019*), and this could be applied to extend the uses of TBGB_SS.

| | | TBGB_AM | TBGB_EwE | TBGB_SS |
|---|---|---|---|---|
| Fishing | Alternative fishing scenarios (removals) | ✓ | ✓ | ✓ |
| | Alternative fishing scenarios (area closures) | ✓ | ✗ | ✗ |
| | Alternative fishing scenarios (by-catch including invertebrates) | ✓ | ✓ | ✗ |
| Climate | Alternative levels of primary production | ✓ | ✓ | ✓ |
| | Alternative sea temperatures | ✓ | ✓ | ✗ |
| | Alternative oceanography | ✓ | ✗ | ✗ |
| Land effects | Coastal erosion | ✓ | ✗ | ✗ |
| | River run-off/storm events | ✓ | ✗ | ✗ |
| | Changes in sedimentation | ✓ | ✗ | ✗ |

parameter values were informed using outputs from the Atlantis model, such as $F$ values, and predator:prey spatial overlap, and using these likely increased the similarities in model dynamics from what we might have seen had the models been developed independently. The comparison between model structure still remains useful, but it is worth noting that without having first developed the Atlantis model, these alternative models would need different means of specifying some of the parameter values and thus could produce different model dynamics.

At the species group level, diets could be compared across all three models, although they reflect slightly different aspects of the modelled dynamics. The TBGB_EwE diets were relatively unchanged from the input diets; although they underwent some adjustments during calibration, they cannot really be considered an output of the model. The size-structured model has far greater emphasis on diets of species when they are very small and only eat phyto- or zoo-plankton, or slightly bigger and eating small pelagic fishes, and these aspects of life history are not included in the other two models. TBGB_AM has species recruited to the model as juveniles into the youngest age-class, but they could be several years old upon entering the model depending on the lifespan of the species, due to the equal sized age-classes. TBGB_EwE doesn't model individuals, and has no age-structure, but diets have been defined to assume the species group biomass pools resemble those corresponding to adult and juveniles combined in the TBGB_AM. TBGB_AM diets are perhaps the most reflective of model dynamics, as they are the result of spatial, temporal, growth, gape-size, life-stage, competition, prey availability dynamics as well as prey preferences. The vertebrate species groups generally had good diet correlation between TBGB_EwE and TBGB_AM.

Despite the good diet correlation between TBGB_EwE and TBGB_AM, the keystone analyses produced rather different rankings. This is interesting as we would expect connectivity and influence to be linked with diets. Some of the vertebrate species groups had almost reverse keystone rankings in the two models. For example, pelagic fish large

and elasmobranch invertivores both had high keystone rankings in one model and low in the other, despite positively correlated diets and very similar trophic levels. It's possible these differences came from modelling species groups as biomass pools with no age or size structure compared to modelling individuals with both age and size structure. It is difficult to predict whether this will influence results when exploring scenarios, but it is worth considering that it could.

The emergent growth curves in the base TBGB_AM and TBGB_SS models need to be considered when using these models. While they are within the correct ballpark of the literature curves (the calibration goal for size-at-age outputs (*Pethybridge et al., 2019*)), the differences could still affect model results, and hence need to be considered. A species group which has larger fish for a given age-class in the model than suggested in the literature will suffer less mortality for a given biomass extraction (e.g. from fishing or predation) and for a given number of these fish, we can expect greater predation pressure on their prey. Conversely, smaller than expected fish will suffer a greater mortality in numbers for a given biomass extraction, and for a given number of these fish, we can expect less predation pressure on their prey. Size-at-age may also affect diets and prey selectivities through gape size limitations.

The tuning of model parameters during calibration focused on parameters relating to growth, mortality and predation. This process did not change parameters from known values in the literature, but rather tuned parameters such that the resulting dynamics came in line with those expected based on the literature. For example, calibrating TBGB_AM to achieve realistic growth curves involved checking prey availability, gape size limitation, competition from other predators, and feeding parameters such as handling time and clearance rates. Similarly, to achieve natural mortality in line with expected M values involved ensuring appropriate predation mortality at each age-class. Appropriate predation mortality involves availability to their predators (including spatial and temporal overlap), gape size limitation, other available prey for the predators, and feeding dynamics of the predators.

Having calibrated the model to produce realistic dynamics that are stable in the absence of pressure such as fishing, perturbing the initial conditions then tested the persistence of these dynamics. The lower trophic level species groups were generally more sensitive to perturbations of the initial conditions. While this corresponds well with higher variability expected at lower trophic levels in nature (*Molinero et al., 2008*), it was more likely caused by how we modelled the lower trophic levels (*McGregor, Fulton & Dunn, 2020*), as these were biomass pools with growth effectively instantaneous, and hence could be more responsive. Scallops and flatfish were exceptions as they were both modelled with age structure, and yet displayed high variability under historical fishing. These species groups crashed under historical fishing in some simulations, and rebounded in others, making the range of possible biomasses very wide. Further exploration into which simulations led to which outcomes could help understand when these fisheries might be under threat of collapse. It could also be interesting to test if there was a connection between the two such that they crashed in the same simulations, or perhaps when one crashed the other survived.

Varying the oceanographic variables through sampling ROMS years or repeating a single ROMS year introduced greater variability in the model results than perturbing the initial conditions. This suggests careful consideration needs to be made into the forcing of the oceanographic variables for this Atlantis model. It may be helpful to extend the ROMS model so a longer timeseries of forcing oceanographic variables can be used. It also raises the question of possible effects from varying temperature and perhaps productivity in TBGB_EwE to compare responses and flow-on effects. All of these aspects will be important to consider for climate change scenarios. Climate change scenarios are often driven through the oceanography, and how we apply this is likely to be influential on results. While environmental effects can be simulated in TBGB_EwE, changes in environmental variables would need to use proxies, such as changes in primary productivity (e.g. *Niiranen et al., 2013*; *Hoover, Pitcher & Christensen, 2013*). Climate change scenarios are less likely to be explored using TBGB_SS due to limited scope for including oceanographic changes in this model, and the limited species groups.

The skill assessment of TBGB_AM with respect to CPUE and survey indices spanned from very negatively correlated in the case of rig, to very positively correlated for school shark. In the case of rig, the sudden drop in survey biomass does not seem to relate to a shift in fishing pressure, so it is possible there is a recruitment driver at play that the model isn't picking up on. It is possible in Atlantis to include recruitment variability as a time series, which could help fit this trend in the hindcast, and could be used to introduce variability when exploring scenarios into the future. Scenarios should explore sensitivities on the species groups with low skill to assess the effect of these groups on the scenario outcomes. This may be especially important for a species that also has a high keystone ranking, as this would suggest they are more influential on the rest of the species groups.

That the diversity responses to historical fishing pressure were similar in TBGB_EwE and TBGB_AM suggests these models have captured similar system-level dynamics. It would be interesting to explore this further by projecting both models into the future with varying catch levels, perhaps focused on subsets of the system. This activity might also help us understand the difference in diversity under fishing for TBGB_SS. The end result of reduced diversity was consistent across all three models, but the absence of an initial increase in diversity in the size-structured model remains a mystery. Typically, size-spectrum indicators relate to size (e.g. mean length, maximum length, proportion of large fish) (*Shin et al., 2005*; *Blanchard et al., 2014*). There do not appear to be examples of calculating a diversity index such as Kempton's Q from size-spectrum models.

As TBGB_AM is the most complete representation of the system out of the three models developed here, there are unlikely to be scenarios that can be explored in one of the alternative models but not in TBGB_AM. There may, however, be scenarios where it makes more sense to explore them in one of the simpler models due to shorter run-times and ease of use, especially if we were to transfer the EwE model to an R version using *Rpath*. We could then load many simulations exploring fishing and climate change into the future, and evaluate system-wide responses. We could use these results to define a suitable subset of the scenarios to run in Atlantis.

## CONCLUSIONS

The analyses presented in this paper describe the development and assessment of three ecosystem models of the Tasman and Golden Bays. The most complex of these models is an end-to-end ecosystem model using the Atlantis framework, that is spatially and temporally resolved, and has the most complete representation of the system, including oceanographic dynamics, light, nutrients, primary production, the foodweb consisting of 51 species groups, fisheries, and feedback loops within these components. The Atlantis model was based on accepted standards in the literature, then assessed for initialisation uncertainty, sensitivity to oceanographic variables, and realistic dynamics and responses to historical fishing. The alternative models were both simplifications of the Atlantis model, and were assessed through comparison with the Atlantis model. As parameters were shared between the models, the models differed in structure rather than parameterisation. The models compared generally well for diets, growth and response to historical fishing. Keystone analyses were not consistent between models, and this should be considered when using these models in parallel. We have shown the Atlantis model to be robust under initialisation uncertainty, except for scallops and flatfish which presented varied responses to historical fishing pressure. The Atlantis model was most sensitive to varying the oceanographic variables, suggesting these sensitivities are important to include when using this model. The alternative models produced similar dynamics to the Atlantis model, and we recommend running these models in parallel where they are suited for exploring scenarios. We recommend the use of proxies to extend the range of possible scenarios for these simpler models. In using these models to explore scenarios into the future, we recommend analysing how each model produces a result before trying to understand what the result can teach us about the ecosystem, and we recommend to always include sensitivities.

## ACKNOWLEDGEMENTS

Mark Hadfield (NIWA) for development of the ROMS model for oceanographic variables. Bec Gorton (CSIRO) for converting the ROMS variables into Atlantis model inputs. James Williams and Keith Micheal (NIWA) for scallops advice. Sean Handley (NIWA) for advice around the bioregionalisation.

### Funding

This work was funded under NIWA project FIFI2001 and Sustainable Seas Phase 1. The funders had no role in study design, data collection and analysis, decision to publish, or preparation of the manuscript.

### Grant Disclosures

The following grant information was disclosed by the authors:
NIWA Project: FIFI2001.
Sustainable Seas Phase 1.

## Competing Interests

Vidette L. McGregor, Adele Dutilloy, Samik Datta, and Ian Tuck are employed by National Institute of Water and Atmospheric Research (NIWA) Ltd. Javier Porobic is employed by the Commonwealth Scientific and Industrial Research Organisation (CSIRO). Alistair Dunn is employed by Ocean Environmental.

## Author Contributions

- Vidette L. McGregor conceived and designed the experiments, performed the experiments, analyzed the data, prepared figures and/or tables, authored or reviewed drafts of the paper, and approved the final draft.
- Peter Horn conceived and designed the experiments, performed the experiments, analyzed the data, prepared figures and/or tables, authored or reviewed drafts of the paper, and approved the final draft.
- Adele Dutilloy conceived and designed the experiments, performed the experiments, analyzed the data, prepared figures and/or tables, authored or reviewed drafts of the paper, and approved the final draft.
- Samik Datta conceived and designed the experiments, performed the experiments, analyzed the data, prepared figures and/or tables, authored or reviewed drafts of the paper, and approved the final draft.
- Alice Rogers conceived and designed the experiments, performed the experiments, analyzed the data, authored or reviewed drafts of the paper, and approved the final draft.
- Javier Porobic performed the experiments, authored or reviewed drafts of the paper, and approved the final draft.
- Alistair Dunn conceived and designed the experiments, authored or reviewed drafts of the paper, and approved the final draft.
- Ian Tuck conceived and designed the experiments, authored or reviewed drafts of the paper, and approved the final draft.

## Data Availability

The data and code are available at GitHub: https://github.com/mcgregorv/TBGB.

## Supplemental Information

Supplemental information for this article can be found online at http://dx.doi.org/10.7717/peerj.11712#supplemental-information.

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
