# Peer review of "From data compilation to model validation: comparing three ecosystem models of the Tasman and Golden Bays, New Zealand"

_PeerJ, doi:10.7717/peerj.11712_

## Round 0.1 · original submission · Major Revisions

Please address the reviewer comments and resubmit.

·

Basic reporting

Page 18 lbe 418 missed links/reference to Table twice.

AM is largely a predatory/prey fisheries model. The limitations in terms of the role that the AM model is to play in ecosystem based management is not clearly defined, although it is presented in the context of broader EBM.

Many useful experiments have been run and results are reported. However, the discussion of what this means is weak.

Conclusion missing.

Experimental design

There is circular thinking in the experimental design. While the three models are independent in principle, parameters from AM were used to populate EWE and SS.

If three models are compared, it would be good to include a table that compares the models, beyond the strengths/weaknesses on particular items.

If this is the best contemporary 'what-if' model then more attention could be given to the actual dynamic structures that are at work. It remais a black box expert model with regard to its dyamic capacity.

Calibration and testing of extremes are described in a useful manner.

Connectivity and Influence (6.3) p22 reports findings but there is no conclusion from this paragraphs either. What do the authors learn from this?

Ecosystem based management (EBM) in the introduction includes multiple activties, but the model is fisheries oriented. Seagrass and algae are narrowly included. Considerable assumptions and adjustments had to be made to prevent crashing the model. These considerations should be clearly listed in Discussion and thereafter the Conclusions.

Validity of the findings

The link between the findings and the discussion is not strong.

The conclusion is missing altogether.

The recommendation in the abstract doesn't clearly line up with discussion, given that conclusion is missing. 'We recommend that scenarios relating to ecosystem dynamics of the TBGB ecosystem incorporate initialisation uncertainty, oceanographic uncertainty, and compare responses across all three models where it is possible to do so."

Additional comments

The process of testing the Atlantis model for this bay is very useful. The comparison with the other models is less evident. When would the authors recommend use the AM model and when does it come with caution?

A summary table of comparision between the models might help.

A conclusion is essential.

Mentioning of the national science challenge Sustainable Seas in the abstract doesn't seem relevant. Perhaps move this to the introduction. While acknowledged that the research is incremental, it seems important to position the contribution and the limitation of this model to tie the introduction and the Discussion (Conclusion?) together.

Reviewer 2 ·

Basic reporting

- Figure 3 can be moved to the Appendix (if it is not of course providing any critical information for the model excluding assumptions/inputs).
- I am not sure how to interpret Figure 4 without looking at the text. Would be nice to provide some complementary explanation in the Figure caption.
- Page 18 has multiple formatting issues. Page 32 has several English issue.

Experimental design

- The abstract is not self-descriptive. What is the aims of this research and what is its contribution and industrial application?
- Page 4, it is important to add some information about the methods used for developing the Atlantis model and if it is too difficult to fit the observation, then how we can rely on the results of scenario?

Validity of the findings

- The goodness of calibration has not been reported. By looking at the literature and the values of model outputs in Figure 5, the error looks quite big. Therefore, without providing this type of statics it is hard to say how good the model is calibrated.
- Page 20, 21, was the sensitivity test global or local? What method has been used in what range?
- The managerial implications of this study should be well-explained according to aforementioned results. What advantage industry and practitioners can gain from the findings of your study? What are the specific action plans based on the research findings? These should be addressed.
- The limitations need to be highlighted. Furthermore, the suggestions for future studies could be carefully expanded.

Additional comments

This study aims to Compare three ecosystem models of the Tasman and Golden Bays. The authors have explained the standard tests of models and compared the results of certain scenarios across three. The results are interesting and can contribute to the literature of ecological modeling. Please see few suggestions and comments.

---

## Round 0.2 · accepted · Accept

Your revisions addressed the reviewer's concerns and the article is now ready for publication.

Reviewer 2 ·

Basic reporting

This article meets all the requested standards.

Experimental design

The authors have addressed all these points in the revised manuscript.

Validity of the findings

To the best of my knowledge the method and results are valid.

Additional comments

Thanks for your responses and clarifications. The text reads very well after the revision.